# BYPASSING THE RANDOM INPUT MIXING IN MIXUP

## ABSTRACT

Mixup and its variants have promoted a surge of interest due to their capability of boosting the accuracy of deep models. For a random sample pair, such approaches generate a set of synthetic samples through interpolating both the inputs and their corresponding one-hot labels. Current methods either interpolate random features from an input pair or learn to mix salient features from the pair. Nevertheless, the former methods can create misleading synthetic samples or remove important features from the given inputs, and the latter strategies incur significant computation cost for selecting descriptive input regions. In this paper, we show that the effort needed for the input mixing can be bypassed. For a given sample pair, averaging the features from the two inputs and then assigning it with a set of soft labels can effectively regularize the training. We empirically show that the proposed approach performs on par with state-of-the-art strategies in terms of predictive accuracy.

## 1 INTRODUCTION

Deep neural networks have demonstrated their profound successes in many challenging real-world applications, including image classification (Krizhevsky et al., 2012), speech recognition (Graves et al., 2013), and machine translation (Bahdanau et al., 2015; Sutskever et al., 2014). One key factor attributing to such successes is the deployment of effective model regularization techniques, which empower the learning to avoid overfitting the training data and to generalize well to unseen samples. This is because current deep models typically embrace high modeling freedom with a very large number of parameters. To this end, many regularizers for deep models have been introduced, including weight decay (Hanson & Pratt, 1988), dropout (Srivastava et al., 2014), stochastic depth (Huang et al., 2016), batch normalization (Ioffe & Szegedy, 2015), and data augmentation schemes (Cubuk et al., 2019; Hendrycks et al., 2020; Inoue, 2018; Lecun et al., 1998; Simard et al., 1998).

Among those effective regularizers, Mixup (Zhang et al., 2018) is a simple and yet effective, data-augmentation based regularizer for enhancing the deep classification models. Through linearly interpolating random input pairs and their training targets in one-hot representation, Mixup generates a set of synthetic examples with soft labels to regularize the training. Such pairwise, label-variant data augmentation techniques (Guo, 2020; Guo et al., 2019; Kim et al., 2020; Li et al.,

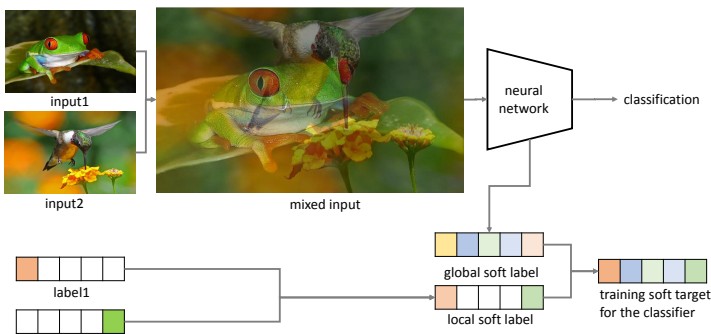

Figure 1: Illustration of the proposed method. The mixed input is the average of the two inputs; the training target for the averaged input is the combination of the local soft label (average of the two one-hot targets) and the global soft label (dynamically generated during training).

2020a; Tokozume et al., 2018a;b; Verma et al., 2019; Yun et al., 2019; Zhang et al., 2018) have attracted a surge of interest and shown their effectiveness on boosting the accuracy of deep networks.

Nevertheless, unlike label-preserving data augmentation such as rotation, flip, and crop, there is still limited knowledge about how to design better mixing policies for sample pairs for effective label-variant regularization. Current Mixup-based approaches either mix a pair of inputs using random mixing coefficients (Guo, 2020; Guo et al., 2019; Summers & Dinneen, 2019; Tokozume et al., 2018b; Verma et al., 2019; Yun et al., 2019; Zhang et al., 2018) or learn to mix salient features from the given pair (Dabouei et al., 2020; Kim et al., 2020; Li et al., 2020b; Walawalkar et al., 2020) to create a set of synthetic samples. Nonetheless, the former methods may create misleading synthetic samples (Guo et al., 2019) or remove important features from the given inputs (Kim et al., 2020) and the latter strategies can incur significant computation cost for identifying and selecting the most descriptive input regions (Kim et al., 2020; Walawalkar et al., 2020).

In this paper, we show that the effort needed for the input mixing with a range of mixing ratios can be bypassed. For a given input pair, one can average the features from the two inputs, and then assign it with a set of soft labels. These soft labels are adaptively learned during training to incorporate class information beyond the provided label pair as well as the evolving states of the training. The method is illustrated in Figure 1, where the mixed input is the pixel-wise average of features from the two inputs and its training target is the combination of the local soft label, which is the average of the two one-hot targets, and the global soft label, which is generated by the networks during training.

We empirically show that the proposed approach performs on par with state-of-the-art methods with random input mixing policy or learning to mixing strategy, in terms of predictive accuracy. We also demonstrate that the synthetic samples created by our method keep tuning the networks long after the training error on the original training set is minimal, encouraging the learning to generate, for each class of the training samples, tight representation clusters. Also, the Class Activation Mapping (CAM) (Zhou et al., 2016) visualization suggests that our method tends to focus on narrower regions of an image for classification.

## 2 BYPASSING THE INPUT MIXING IN MIXUP

### 2.1 MIXUP-BASED DATA AUGMENTATION

For a standard classification setting with a training data set $(X; Y)$, the objective of the task is to develop a classifier which assigns every *input* $x \in X$ a *label* $y \in Y$. Instead of using the provided training set $(X; Y)$, Mixup (Zhang et al., 2018) generates synthetic samples with soft labels for training. For a pair of random training samples $(x^i; y^i)$ and $(x^j; y^j)$, where $x$ is the input and $y$ the one-hot encoding of the corresponding class, Mixup creates a synthetic sample as follows.

$$\widetilde{x}_\lambda^{ij} = \lambda x^i + (1 - \lambda)x^j, \tag{1}$$

$$\widetilde{y}_\lambda^{ij} = \lambda y^i + (1 - \lambda)y^j, \tag{2}$$

where $\lambda$ is a scalar mixing policy for mixing both the inputs and the modeling targets of the sample pair. $\lambda$ is sampled from a Beta$(\alpha, \alpha)$ distribution with a hyper-parameter $\alpha$. The generated samples $(\widetilde{x}_\lambda^{ij}, \widetilde{y}_\lambda^{ij})$ are then fed into the model for training to minimize the cross-entropy loss function.

Current variants of Mixup focus on the introduction of a representation function $\psi$ for input mixing:

$$\widetilde{x}_\lambda^{ij} = \psi(x^i|x^j, \lambda) + \psi(x^j|x^i, 1 - \lambda). \tag{3}$$

The state-of-the-art *random input mixing* variant CutMix (Yun et al., 2019) defines the $\psi$ function to form a binary rectangular mask applying to a randomly chosen rectangle covering $\lambda$ proportion of the input image. PuzzleMix (Kim et al., 2020) is the state-of-the-art *learning to mixing* variant. This method defines the $\psi$ function to compute the saliency map of the input pair, find the optimal mask, and optimize the transport plans for generating the mixed example. This is to ensure that the mixed image contain sufficient target class information corresponding to the mixing ratio $\lambda$ while preserving the local statistics of each input.

### 2.2 FORMING THE TRAINING INPUT

Similar to Mixup and its variants, our method also creates mixed samples from a random sample pair, but for a given sample pair our method does not generate a set of inputs with different features. Instead, for a range of mixing ratios $\lambda$s, our approach uses the same mixed input, invariant to the

mixing ratios provided. We term our method Label driven Mixup (denoted as LaMix). To this end, the mixed input $\widetilde{x}^{ij}$ in LaMix has two forms. For the *random input mixing* methods such as Mixup and CutMix, it is formed by pixel-wise average of the raw features of the input pair $(x^i, x^j)$:

$$\widetilde{x}^{ij} = 0.5x^i + 0.5x^j. \tag{4}$$

For the *learning to mixing* methods such as PuzzleMix, the mixed input is formed by pixel-wise average of the saliency features computed with $\lambda$ value as 0.5:

$$\widetilde{x}^{ij} = \psi(x^i|x^j, \lambda = 0.5) + \psi(x^j|x^i, \lambda = 0.5). \tag{5}$$

The newly resulting input $\widetilde{x}^{ij}$ is then used for training, by passing through a neural network model $f_\varphi$ (parameterized with $\varphi$) to generate the $m$-dimensional input embedding $S^{ij} \in R^m$:

$$S^{ij} = f_\varphi(\widetilde{x}^{ij}). \tag{6}$$

Next, the resulting $S^{ij}$ is then fed into a linear fullyconnected layer $W_l \in R^{c \times m}$ to produce the predicted classification distribution over the $c$ classification target classes:

$$\overline{\overline{y}}^{ij} = \text{Softmax}(W_l S^{ij}). \tag{7}$$

The error of the training is computed by comparing the prediction $\overline{\overline{y}}^{ij}$ and the training target of the input $\widetilde{x}^{ij}$, which is discussed next.

## 2.3 FORMING THE TRAINING TARGET

For the same mixed input $\widetilde{x}^{ij}$ created from a random input pair ($x^i, x^j$), LaMix associates it with various soft target label $\ddot{y}^{ij}$, which is learned by the networks with two steps: 1) obtaining global soft label and 2) integrating with local soft label. In detail, the dynamically assigned soft target label $\ddot{y}^{ij}$ is a function of $(x^i, x^j, y^i, y^j, \lambda)$, as follows.

$$\ddot{y}^{ij}_\lambda = \tau(x^i, x^j, \lambda, y^i, y^j). \tag{8}$$

As illustrated in Figure 1, LaMix implements the $\tau(\cdot)$ function through adding an additional fullyconnected layer $W_t \in R^{c \times m}$ to the original networks. That is, for the given input $\widetilde{x}^{ij}$ with $\lambda$, LaMix first computes its probabilities over the $c$ classification targets, denoted as $p^{ij}_\lambda$:

$$p^{ij}_\lambda = \text{Softmax}(\sigma(W_t S^{ij})), \tag{9}$$

where $\sigma$ denotes the Sigmoid function, and $S^{ij}$ is the same input embedding as that in Equation 7. In other words, the two predictions (i.e., Equations 7 and 9) share the same networks except for the last layer. Because the predictions $p^{ij}_\lambda$ are about all the $c$ target classes of the given data, such predictions thus reflect the class information beyond the provided local label pair, namely $(y^i, y^j)$. For description purpose, we term $p^{ij}_\lambda$ global soft label. The Sigmoid function here will provide the probability of associating the input to a particular label.

After having the global soft label $p^{ij}_\lambda$, we then integrate it with the local label of the given sample pair, i.e., $\widetilde{y}^{ij}_\lambda$ as described in Equation 2, as follows:

$$\ddot{y}^{ij}_\lambda = \beta\widetilde{y}^{ij}_\lambda + (1 - \beta)p^{ij}_\lambda, \tag{10}$$

where $\beta$ is a scalar coefficient between [0,1]. Doing so, we enable the synthetic sample $(\widetilde{x}^{ij}, \ddot{y}^{ij}_\lambda)$ to have information from both the local labels of the provided sample pair (i.e., $\widetilde{y}^{ij}_\lambda$) and the class information beyond the labels of the given pair (i.e., $p^{ij}_\lambda$). This process is also illustrated in Figure 1.

## 2.4 PARAMETER OPTIMIZATION

For training, LaMix minimize, with gradient descent, the cross entropy loss $E$ between the assigned training soft target $\ddot{y}^{ij}_\lambda$ and the network's predicted classification distribution $\overline{\overline{y}}^{ij}$:

$$E = \ddot{y}^{ij}_\lambda log\overline{\overline{y}}^{ij} \tag{11}$$

During training, we also need to prevent LaMix from assigning target soft labels too far away from the gold labels in the original training set. To attain this goal, we alternatively feed samples to the networks with either a mini-batch with original inputs, i.e., $\widetilde{x}^{ii}$, or a mini-batch from the mixed inputs, i.e., $\widetilde{x}^{ij}$. Note that, when training with the former, the networks still need to learn to assign the training target $\ddot{y}^{ii}_\lambda$ to the sample $\widetilde{x}^{ii}$.

|  |  | MNIST | Fashion | SVHN | Cifar10 | Cifar100 |
|---|---|---|---|---|---|---|
| PreAct ResNet-18 | Vanilla | 0.62±0.05 | 4.78±0.19 | 3.64±0.42 | 5.19±0.30 | 24.19±1.27 |
|  | Mixup | 0.56±0.01 | 4.18±0.02 | 3.37±0.49 | 3.88±0.32 | 21.10±0.21 |
|  | CutMix | 0.59±0.06 | 4.06±0.06 | 2.82±0.22 | 4.09± 0.44 | 19.77±0.26 |
|  | LaMix | **0.50±0.04** | **4.03±0.09** | **2.69±0.08** | **3.82±0.18** | **19.38±0.16** |
| ResNet-50 | Vanilla | 0.61±0.05 | 4.55±0.14 | 3.22±0.05 | 4.83±0.30 | 23.10±0.62 |
|  | Mixup | 0.57±0.03 | 4.31±0.05 | 2.85±0.07 | 4.29±0.28 | 19.48±0.48 |
|  | CutMix | 0.49±0.03 | 4.11±0.11 | 2.61±0.08 | 4.57±0.31 | 18.98±0.37 |
|  | LaMix | **0.47±0.04** | **4.10±0.07** | **2.39±0.06** | **3.60±0.24** | **18.60±0.69** |

Table 1: Error rate (%) of methods with PreAct ResNet-18 and ResNet-50 as baselines. We report mean scores over 5 runs with standard deviations (denoted ±). Best results highlighted in **Bold**.

## 3 EXPERIMENTS

We compare our method with both *random input mixing* and *learning to mixing* Mixup variants.

### 3.1 COMPARE WITH RANDOM INPUT MIXING METHODS

We use five image classification tasks. **MNIST** is a digit (1-10) recognition dataset with 60,000 training and 10,000 test gray-level, 28x28-dimensional images. **Fashion** is an image recognition dataset having the same scale as MNIST, containing 10 classes of fashion product pictures. **SVHN** is the Google street view house numbers recognition data set with 73,257 digits, 32x32 color images for training, 26,032 for testing, and 531,131 additional, easier samples. We did not use the additional images. **Cifar10** is an image classification task with 10 classes with 50,000 training and 10,000 test samples. **Cifar100** is similar to Cifar10 but with 100 classes and 600 images each.

We experimented using networks PreAct ResNet-18 (He et al., 2016) and ResNet-50 (He et al., 2016) (denoted as Vanilla model). To fit the MNIST and Fashion data into the same convolutional network architecture, we just simply resize the 28x28 image to 32x32 as that of Cifar10, Cifar100, and SVHN. Each reported value (accuracy or error rate) is the mean of five runs, on a NVIDIA GTX TitanX GPU with 12GB memory. The $\beta$ as discussed in Equation 10 was set as 0.5 unless otherwise specified.

We compare with Mixup (Zhang et al., 2018) and the state-of-the-art *random input mixing* method CutMix (Yun et al., 2019). For Mixup, we use the code from Facebook at [1]. For CutMix, we use the authors' implementation in [2]. For LaMix, the added fully connected layer is the same as the fullyconnected layer of the original network with a Softmax function on the top. For all the comparison baselines, the mixing ratio $\lambda$ is sampled uniformly between [0,1]. All models are trained using mini-batched (128 examples) backprop, as specified in (Zhang et al., 2018) and with the exact settings as in the Facebook codes, for 400 epochs.

#### 3.1.1 PREDICTIVE ACCURACY

The predictive error rates obtained by the Vanilla, Mixup, CutMix, and LaMix with PreAct ResNet-18 and ResNet-50 as baselines are in Table 1.

Table 1 shows that LaMix performed on par with or outperformed the comparison models on all the five datasets. Results also indicate that the performance of CutMix and Mixup was sensitive to the datasets used: CutMix outperformed Mixup on some cases, but loss on some other datasets. This may due to the noisy mixed samples these two methods generated during training. For example, on Cifar10 and MNIST with PreAct ResNet-18 or ResNet-50, CutMix obtained lower accuracy than Mixup. But

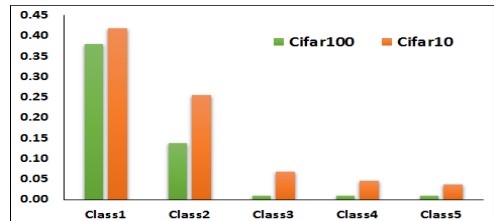

Figure 2: Average values (y-axis) of the top five soft targets (x-axis) used by LaMix for training on Cifar100 (green bars) and Cifar10 (red bars).

on Cifar100 and SVHN with PreAct ResNet-18, CutMix outperformed Mixup with a large margin. In contrast, LaMix performed consistently better than CutMix and Mixup on all the cases.

These results suggest that, although using the same mixed image for a wide range of mixing ratios, LaMix is able to assign a set of appropriate soft labels to the same image, so that they can effectively

---

[1]https://github.com/facebookresearch/mixup-cifar10
[2]https://github.com/clovaai/CutMix-PyTorch

regularize the training to achieve regularization effect as effective as that of *random input mixing* methods, which rely on mixing different features from a given image pair to form a set of images.

### 3.1.2 EVOLVING SOFT TRAINING TARGETS

In Figure 2, we visualize the top five largest training targets used by LaMix. We provide the average values of the top targets during training for both Cifar100 (left) and Cifar10 (right). Results in Figure 2 indicate that LaMix re-distributes a large portion of the one-hot distribution to the other classes. For both Cifar100 and Cifar10, half of the one-hot dis-

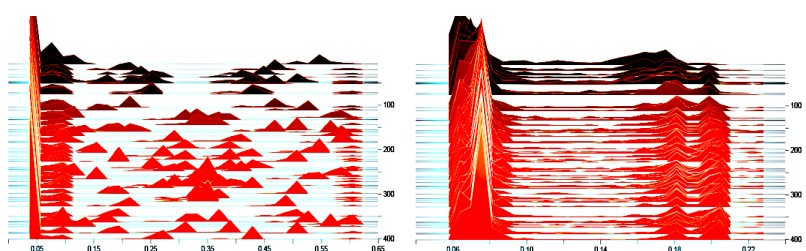

Figure 3: Soft targets used for training (left subfigure) and the global soft labels generated (right subfigure) for a batch of training samples over 400 epochs. The x-axis depicts the soft target values and the y-axis is the distribution mass over the 400 training epochs (from top to down).

tribution mass is re-assigned to the other targets. For example, as shown in Figure 2, the second largest targets for the Cifar100 and Cifar10 are 0.14 and 0.25, respectively.

In Figure 3, we also visualize the assigned soft labels of a random mini-batch of training samples in LaMix with PreAct ResNet-18 on Cifar10 across the 400 epochs of training. The left depicts the soft targets used for training by LaMix, and the right shows the learned global soft labels. In these figures, the x-axis depicts the soft target values and the y-axis is the distribution mass over different training epochs (from top to bottom).

Results in the left subfigure of Figure 3 indicate that the soft target labels assigned to the same batch of training samples keep changing during the training over the 400 epochs (from top to bottom), with the largest soft target training signals laying between 0.55 and 0.65. The right subfigure suggests that the

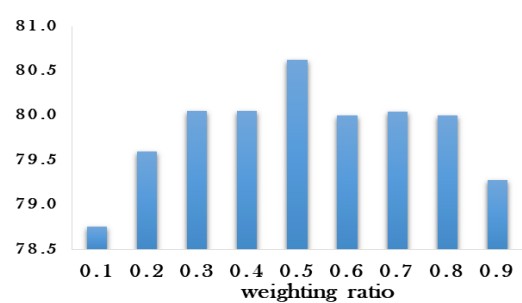

Figure 4: Accuracy (%) obtained by LaMix while varying the weight ratio $\beta$ on Cifar100.

learned global targets for the batch of training samples tend to have small values, containing a large group with values between 0.14 to 0.18, during the early phases of training (top) and then having more values laying between 0.18 and 0.20 as the training progresses (bottom of the figure).

These results imply that for the same mixed image, the LaMix is able to assign it with different soft targets, thus increasing the size of the training set but bypassing the efforts needed for input mixing.

### 3.1.3 RE-WEIGHT THE LOCAL AND GLOBAL SOFT LABELS

We here evaluate the impact of different weight factors for $\beta$ as discussed in Equation 10 for the two soft labels, by varying it from 0.1 to 0.9. The results obtained by LaMix using PreAct ResNet-18 on Cifar100 are in Figure 4.

The results in Figure 4 suggest that averaging the two sets of soft labels provides better accuracy than other weighting ratios for this dataset. Also, the

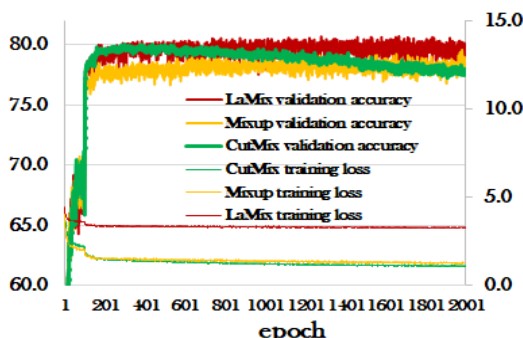

Figure 5: Training loss (right y-axis) and validation accuracy (left y-axis) of LaMix, Mixup, and CutMix across 2K training epochs.

LaMix obtained similar accuracy when the $\beta$ fell into the region between [0.3, 0.8].

| Network Architecture | Vanilla | Mixup | CutMix | LaMix |
|---|---|---|---|---|
| LeNet | 58.9±2.88 | 56.4±1.42 | 62.5±3.09 | **56.3±1.82** |
| WideResNet-28-10 | 19.9±0.41 | 17.8±0.31 | 17.9±0.30 | **17.2±0.26** |
| DenseNet-121 | 19.5±0.34 | 17.3±0.13 | 17.3±0.52 | **16.9±0.09** |

Table 2: Error rate (%) of LaMix, Mixup, and CutMix with LeNet, WideResNet28-10 and DenseNet-121 on Cifar100. We report mean scores over 5 runs with standard deviations (denoted ±).

### 3.1.4 TRAINING CHARACTERISTICS

In Figure 5, we plot the training loss and validation accuracy across an extended training period, namely 2k training epochs, of LaMix, Mixup, and CutMix with PreAct ResNet-18 on Cifar100.

Figure 5 shows that the training loss of LaMix (thin red curve) maintains a relatively higher level than the other two methods, allowing the model to keep tuning the networks. Promisingly, as shown by the accuracy curve (wide red), even training for a long time, LaMix is not overfitting the training set.

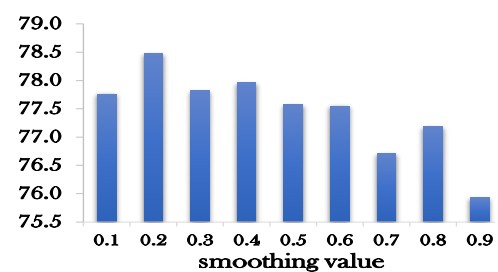

Figure 6: Accuracy (%) of label smoothing with various smoothing factors on PreAct ResNet-18.

### 3.1.5 PERFORMANCE ON SHALLOWER, DEEPER, AND WIDER NETWORKS

We also test LaMix using different network architectures on Cifar100, including a shallow network LeNet (Lecun et al., 1998), a wider network WideResNet-28-10 (Zagoruyko & Komodakis, 2016), and the 121 layers deep network DenseNet-121 (Huang et al., 2017). For these networks, we use the implementation from Facebook [3].

Similar to that of using PreAct ResNet-18 and ResNet-50, LaMix performed consistently on par with or superior to the comparison models on these three additional network architectures.

### 3.1.6 COMPARISON WITH LABEL SMOOTHING TECHNIQUE

We also compare our method LaMix with label smoothing techniques as presented in (Müller et al., 2019; Pereyra et al., 2017; Szegedy et al., 2016). These techniques train models with soft targets that are a weighted average of the hard targets and the uniform distribution over labels with a smoothing factor, aiming at preventing the network from becoming over-confident. We varies the smoothing factor from 0.1 to 0.9 on PreAct ResNet-18 for Cifar100. Accuracy results are presented in Figure 6.

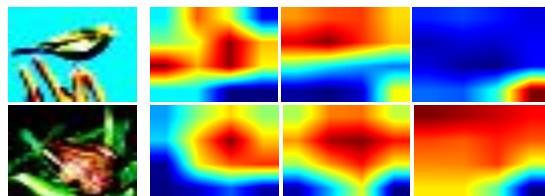

Figure 7: Left: two classes of Cifar10 testing images; The right three depict the visualized CAMs of LaMix, Mixup, and CutMix, respectively.

Results in Figure 6 show that label smoothing can improve the predictive accuracy of PreAct ResNet-18 if the smoothing factor being tuned properly, but even the best accuracy obtained is still far below that achieved by LaMix. For example, the best accuracy 78.47% obtained by PreAct ResNet-18 is with label smoothing factor of 0.2, which is still below the 80.62% achieved by LaMix.

### 3.1.7 CLASS ACTIVATION MAPPING (CAM) VISUALIZATION

After the 400 training epochs, we, in Figure 7, depict the CAM maps (Zhou et al., 2016) of two classes (*bird* and *frog*) using testing samples from Cifar10. The test images are on the left, and the

---

[3]https://github.com/facebookresearch/mixup-cifar10/blob/master/models/

visualization CAMs created by LaMix, Mixup, and CutMix are depicted on the second left, second right, and the right, respectively. One can see from Figure 7 that, LaMix tends to focus on narrower discriminative regions of the images than Mixup and CutMix.

### 3.1.8 REGULARIZATION EFFECT OF THE SYNTHETIC SAMPLES

We also visualize the impact of the synthetic samples created by LaMix. We present the results of three classes (i.e., Cat, Dog, and Deer) in Cifar10 with a 2D bottleneck hidden representation in the middle of the network. In specific, following the 512 filters generated by the PreAct ResNet-18 (namely the layer before the Softmax layer), we use a fully-connected Tanh layer of 2 units as a bottleneck layer, followed by two fully-connected Tanh layers with 100 and 512 units respectively. We visualize the 2D bottleneck representation for the original input samples at the $100^{th}$, $400^{th}$, $700^{th}$, and $1k^{th}$ training epoch. For the last three, the training errors were 100%. Figure 8 depicts the results, with different colors for the three classes.

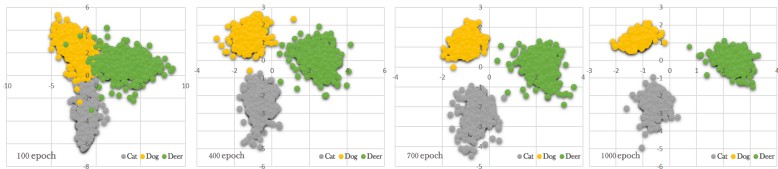

Figure 8: 2D data embeddings of three classes in Cifar10 at the $100^{th}$, $400^{th}$, $700^{th}$, and $1k^{th}$ epoch.

Results in Figure 8 show that, as the learning progresses, the embeddings for the training samples are tuned gradually towards separating the three classes and the model achieves 100% accuracy at the $400^{th}$ epoch (second left). On the right two subfigures (with training accuracy of 100%), the network keeps tuning using the newly created samples. At the $1000^{th}$ epoch (right), the input embeddings are separated into three tight clusters with distance far away than that of the $400^{th}$ epoch (second left). These results suggest that the synthetic samples help LaMix create tight representation cluster for training samples in each class and widen the gap between these clusters. These turnings happened long after the training accuracy on the original training set was not able to further increase.

## 3.2 COMPARE WITH LEARNING TO MIXING METHOD

In this section, we compared LaMix with the state-of-the-art *learning to mixing* variant of Mixup, namely the PuzzleMix (Kim et al., 2020). We use the authors' code published at [4]. We trained the networks for 600 epochs as implemented in the provided source codes, and kept all the settings in the codes. Each reported error is the mean of five runs, on a NVIDIA GTX TitanX GPU with 12GB memory. In this set of experiments, the $\beta$ in LaMix was set as 1.0 unless otherwise specified.

| | | MNIST | Fashion | SVHN | Cifar10 | Cifar100 |
|---|---|---|---|---|---|---|
| | Vanilla | 0.62±0.05 | 4.78±0.19 | 3.64±0.42 | 5.19±0.30 | 24.19±1.27 |
| PreAct ResNet-18 | Puzzle | 0.57±0.04 | 4.18±0.06 | 2.80±0.04 | 3.16±0.09 | 19.56±0.30 |
| | LaMix | 0.56±0.14 | 4.12±0.04 | 2.92±0.09 | 3.17±0.10 | 19.91±0.28 |
| | Vanilla | 0.61±0.05 | 4.55±0.14 | 3.22±0.05 | 4.83±0.30 | 23.10±0.62 |
| ResNet-50 | Puzzle | 0.49±0.02 | 3.99±0.07 | 2.44±0.04 | 2.55±0.13 | 16.75±0.22 |
| | LaMix | 0.51±0.02 | 3.99±0.05 | 2.33±0.04 | 2.80±0.02 | 16.69±0.17 |

Table 3: Error rate (%) of the testing methods with PreAct ResNet-18 and ResNet-50 as baselines. We report mean scores over 5 runs with standard deviations (denoted ±).

Results on the five testing datasets as used to evaluate the *random input mixing* methods in Section 3.1 are presented in Tables 3. Tables 3 shows that LaMix performed on par with the PuzzleMix in terms of accuracy obtained. As shown in the table, LaMix obtained very similar accuracy on all the testing cases as the PuzzleMix method, except for the ResNet-50 on the Cifar10 dataset, which PuzzleMix slightly outperformed LaMix with 0.25%. These results suggest that mixing the input features with a range of mixing ratios $\lambda$s in PuzzleMix can be bypassed without degrading the predictive accuracy.

---

[4]https://github.com/snu-mllab/PuzzleMix

### 3.2.1 EXPERIMENTS ON TINY-IMAGENET

The authors' codes for PuzzleMix also include settings for running PuzzleMix on Tiny-ImageNet (Chrabaszcz et al., 2017), so here we also evaluated LaMix using this dataset. This dataset has 200 classes, each with 500 training and 50 test 64x64x3 images. We presented the results in Table 4.

| Tiny-ImageNet | | |
|---|---|---|
| PreAct ResNet-18 | PuzzleMix | 36.38±0.23 |
| | LaMix | 36.32±0.36 |
| ResNet-50 | PuzzleMix | 30.73±0.22 |
| | LaMix | 30.68±0.24 |

Table 4: Error rate (%) on Tiny-ImageNet.

Results in Table 4 show that LaMix obtained similar error rates when compared with PuzzleMix on this dataset. For both testing network architectures, namely PreAct ResNet-18 and ResNet-50, the accuracy difference between LaMix and PuzzleMix is marginal. These results further indicate that the input mixing effort with a range of mixing ratios $\lambda$s in the *learning to mixing* variant PuzzleMix can be bypassed to conduct effective model regularization.

## 4 RELATED WORK

Various variants of the Mixup (Zhang et al., 2018) method have been introduced, devoting to creating a set of mixed inputs by either randomly mixing the inputs (Guo, 2020; Guo et al., 2019; Tokozume et al., 2018a;b; Verma et al., 2019; Yun et al., 2019; Zhang et al., 2018) or learning to mixing the informative features from an input pair (Dabouei et al., 2020; Kim et al., 2020; Li et al., 2020a;b; Walawalkar et al., 2020). In the former category, Mixup (Zhang et al., 2018) mixes all the input features with a random mixing ratio. CutMix (Yun et al., 2019) creates a mixed input by copying a box of pixels from one image and pasted it to the other image. The second family of approaches aim to learn to mixing important input features (Dabouei et al., 2020; Kim et al., 2020; Li et al., 2020a;b; Walawalkar et al., 2020). For example, the recent method PuzzleMix (Kim et al., 2020) leverages the optimal transport to combine the identified salient features from the inputs. There is also a method called Manifold Mixup (Verma et al., 2019), which conducts feature interpolation in addition to mixing the input pair. This approach performs feature mixing on a randomly selected layer of the networks. Unlike these methods which generate a set of mixed inputs from a sample pair, our strategy creates only one mixed input by averaging the features from a given sample pair.

Our method also relates to the label smoothing techniques, which regularize the output distribution of neural models to penalize high-confidence softmax distributions (Müller et al., 2019; Pereyra et al., 2017; Szegedy et al., 2016). These techniques train models with soft targets that are a weighted average of the hard targets and the uniform distribution over labels with a smoothing factor. These smoothing methods, however, do not relate the smooth labels with the associated features but utilize uniform label smoothing. Our approach learns to form soft targets based on the features of the input pair and the states of the evolving networks.

The generation of the global soft labels in LaMix is also related to how the training targets generated by self-distillation methods (Ahn et al., 2019; Furlanello et al., 2018; Hinton et al., 2015; Mobahi et al., 2020; Yang et al., 2019). However, self-distillation treats the final predictions as target labels for a new round of training, and the teacher and student architectures are identical (Mobahi et al., 2020). In LaMix, the classifier network and the network for the global soft label have different architectures, and the training targets for the classification model are augmented by the global soft labels.

## 5 CONCLUSION AND FUTURE WORK

We contributed a finding that one can bypass the effort needed for input mixing with a range of mixing ratios in Mixup-based approaches to conduct effective pairwise, label-variant model regularization. Consequently, one can mitigate creating noisy synthetic samples from the given inputs or reduce the computation cost required for selecting salient regions from the inputs.

Our observation may shed light on how one may conduct label-variant data augmentation in domains that are expensive to perform input mixing with a range of mixing ratios.

# 6 ADDITIONAL EXPERIMENTS

## 6.1 MIXUP PLUS LABEL SMOOTHING

We conducted experiments of stacking uniform label smoothing (with coefficients of 0.1, 0.2, 0.3 and 0.4) on top of Mixup, using PreAct ResNet-18 and ResNet-50 on Tiny ImageNet, Cifar100 and Cifar10. We presented the results in Table 5.

| Dataset | network | LaMix | Mixup | Mixup+ULS0.1 | Mixup+ULS0.2 | Mixup+ULS0.3 | Mixup+ULS0.4 |
|---------|---------|-------|-------|--------------|--------------|--------------|--------------|
| TinyImageNet | ResNet18 | 34.40±0.26 | 41.68±0.35 | 42.31 ±0.34 | 45.19±0.68 | 60.58±0.59 | 78.48±0.53 |
| | ResNet50 | 28.85±0.15 | 39.09±0.28 | 42.58 ±0.39 | 59.25±0.91 | 81.06±0.24 | 91.82±0.37 |
| Cifar100 | ResNet18 | 19.38±0.16 | 21.10±0.21 | 21.51±0.51 | 21.41±0.55 | 20.94±0.49 | 20.95±0.49 |
| | ResNet50 | 18.60±0.69 | 19.48±0.48 | 21.58 ±0.86 | 20.87±0.51 | 21.64±0.41 | 21.18 ±0.58 |
| Cifar10 | ResNet18 | 3.82±0.18 | 3.88±0.32 | 4.00±0.17 | 3.95±0.13 | 4.06±0.04 | 4.06±0.03 |
| | ResNet50 | 3.60±0.24 | 4.29±0.28 | 4.02±0.27 | 4.09±0.10 | 4.19±0.18 | 4.68±0.79 |

Table 5: Error rate (%) of the testing methods with PreAct ResNet-18 and ResNet-50 as baselines. We report mean scores over 5 runs with standard deviations (denoted ±).

These results show that label smoothing degraded the accuracy of vanilla Mixup in the cases of Cifar100, Cifar10, and Tiny ImageNet. Such accuracy degradation is due to the two layers of label smoothing in Mixup. Research has shown that the mixing label in the original Mixup has the effect of label smoothing (Carratino et al., 2020). In this sense, adding another label smoothing regularizer on top of the vanilla Mixup means that the Mixup will have two layers of label smoothing regularizer, which may mess up the regularization effect, thus degrading the accuracy as shown in the results above.

## 6.2 RANDOM INPUT MIXING IN TINY IMAGENET

We conducted additional experiments of random input mixing on Tiny ImageNet, and present the results in Table 6. Similar to that in Table 1, results here again suggest that, although using the same mixed image for a wide range of mixing ratios, LaMix is able to assign a set of appropriate soft labels to the same image, so that they can effectively regularize the training to achieve regularization effect as effective as that of random input mixing methods, which rely on mixing different features from a given image pair to form a set of images.

| Tiny ImageNet | Vanilla | Mixup | CutMix | LaMix |
|---------------|---------|-------|--------|-------|
| PreAct ResNet18 | 42.61±0.41 | 41.68±0.35 | 43.01 ±0.36 | 34.40±0.26 |
| ResNet50 | 40.89±0.36 | 39.09±0.28 | 41.36±0.44 | 28.85±0.15 |

Table 6: Error rate (%) of the testing methods with PreAct ResNet-18 and ResNet-50 as baselines on Tiny ImageNet. We report mean scores over 5 runs with standard deviations (denoted ±).

## 6.3 WITHOUT ORIGINAL TRAINING SAMPLES

Without the original samples, the LaMix method significantly degraded its accuracy. We conducted the experiments and provided the results in Table 7.

| Dataset | Network | LaMix | LaMix w/o original samples |
|---------|---------|-------|----------------------------|
| Cifar100 | PreAct ResNet18 | 19.38 ±0.16 | 25.74±0.43 |
| | ResNet50 | 18.60±0.69 | 25.57±0.36 |
| Cifar10 | PreAct ResNet18 | 3.82±0.18 | 5.90±0.31 |
| | ResNet50 | 3.60±0.24 | 6.39±0.59 |

Table 7: Error rate (%) of the testing methods with PreAct ResNet-18 and ResNet-50 as baselines. We report mean scores over 5 runs with standard deviations (denoted ±).

The reason here is that, without the original training samples, the training samples could be very different from the testing samples. Also, it is interesting to see that ResNet50 obtained inferior results than PreAct ResNet18 in some cases when removing the original training samples. This indeed suggests that Resnet50 fits the mixed training samples better due to its expressive power, but those

mixed training samples may be far from the testing samples due to the lack of the original training samples.

## 6.4 WITHOUT GLOBAL SOFT LABEL

Without the global soft label, namely $p^{ij}$, the LaMix model significantly degraded the accuracy. We conducted the experiments and provided the results in Table 8

| Dataset | Network | LaMix | LaMix w/o global soft label |
|---------|---------|-------|------------------------------|
| Cifar100 | PreAct ResNet10 | 19.38±0.16 | 23.03±0.35 |
| | ResNet50 | 18.60±0.69 | 21.34±0.70 |
| Cifar10 | PreAct ResNet10 | 3.82±0.18 | 5.07± 0.38 |
| | ResNet50 | 3.60±0.24 | 4.78±0.14 |

Table 8: Error rate (%) of the testing methods with PreAct ResNet-18 and ResNet-50 as baselines. We report mean scores over 5 runs with standard deviations (denoted ±).

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
