# OpenReview forum: "Bypassing the Random Input Mixing in Mixup"
_ICLR.cc/2021/Conference — Reject_

### Official Review · AnonReviewer2 · 2020-10-15
**Simple combination of mixup and self-distillation, but marginal experimental results**

**Rating:** 5
**Confidence:** 3

**Review:**

Summary

This paper simply combines mixup and self-distillation to achieve more adaptive soft label, which effectively regularize the training. In the manuscript, authors argue that the existed mixup-based approaches has two mainly efforts, may create misleading training samples or meet computation cost issue on creating samples. Motivated by this, they propose "LaMix", which can leverage the information of self-distillation, to solve those two efforts and achieve competitive performance with SOTA "Puzzle-mixup".


Comment

1. This paper introduces a combination method between mixup and self-distillation, and simply use an additional FC layer to have adaptive soft label, which is intuitive and clear but lack of novelty. Can you give more insightful comment about how adaptive label can help mixup soft label?

2. In Figure 3, I think the provided evidence for the effect of considering adaptive soft label with mixup approach is promising.

3. For experimental results, in section 3.2, the proposed "Lamix" seems not achieve significant improvement compared to the SOTA "Puzzle-mix" on CIFAR-10 and 100.

---

> ### Author Response · Authors · 2020-11-21
> **Adaptive soft label mitigates the label mis-matching issue in Mixup and also makes good use of the only mixed input of a sample pair**
>
> Thank you for your insightful comments.
>
> Our work is inspired by Mixup and self-distillation. Our method differs from Mixup in that we average the two inputs, instead of linearly interpolating the inputs with a random ratio. In other words, the Mixup method will generate a large set of mixed inputs from a given sample pair, but our approach only has one mixed input (by average).
>
> Regarding your comments:
>
> 1. Our work is motivated by two issues in Mixup. First, there is a dis-matching between mixed input and associated label of a synthetic sample in Mixup during to its random input mixing policy, thus Mixup can generate misleading synthetic samples. Second, the input mixing for learning to mixing Mixup strategies incurs significant computation cost for selecting descriptive input regions. To cope with these two problems in Mixup, our method leverages only one mixed input (by averaging the input pair), and then assigns adaptive soft labels to the mixed input. Dynamically learning a soft label will mitigate the label dis-matching issue in Mixup and also effectively make good use of the only mixed input.
>
>  Simply stacking label smoothing on top of vanilla Mixup didn’t help. We conducted experiments of stacking uniform label
>  smoothing (with different coefficients) on top of Mixup, and here are the results (error rates) of using PreAct ResNet-18 and
>  ResNet-50 on Tiny ImageNet, Cifar100 and Cifar10 (number in the bracket is the deviation of 5 runs).
>
>  Tiny ImageNet:
>
>  LaMix, Mixup,  Mixup + ULS 0.1, Mixup + ULS 0.2, Mixup + ULS 0.3, Mixup + ULS 0.4
>
>  PreAct ResNet18, 34.40 (0.26), 41.68 (0.35),  42.31 (0.34), 45.19 (0.68), 60.58 (0.59), 78.48 (0.53)
>
>  ResNet50, 	 28.85 (0.15), 39.09 (0.28), 42.58 (0.39), 59.25 (0.91), 81.06 (0.24), 91.82 (0.37)
>
>  Cifar100:
>
>  LaMix, Mixup,  Mixup + ULS 0.1, Mixup + ULS 0.2, Mixup + ULS 0.3, Mixup + ULS 0.4
>
>  PreAct ResNet18, 19.38 (0.16), 21.10 (0.21), 21.51 (0.51), 21.41 (0.55), 20.94 (0.49), 20.95 (0.49)
>
>  ResNet50, 	 18.60 (0.69), 19.48 (0.48), 21.58 (0.86), 20.87 (0.51), 21.64 (0.41), 21.18 (0.58)
>
>  Cifar10:
>
>  LaMix, Mixup, Mixup + ULS 0.1, Mixup + ULS 0.2, Mixup + ULS 0.3, Mixup + ULS 0.4
>
>  PreAct ResNet18, 3.82 (0.18), 3.88 (0.32), 4.00 (0.17), 3.95 (0.13), 4.06 (0.04), 4.06 (0.03)
>
>  ResNet50, 	 3.60 (0.24), 4.29 (0.28), 4.02 (0.27), 4.09 (0.10), 4.19 (0.18), 4.68 (0.79)
>
>  These results show that label smoothing degraded the accuracy of Mixup in the cases of Cifar100, Cifar10, and Tiny ImageNet on
>  PreAct ResNet18 and Resnet50. Such accuracy degradation is due to the two layers of label smoothing in Mixup. Research has
>  shown that the mixing label in the original Mixup has the effect of label smoothing (Carratino et al, 2020, On Mixup
>  Regularization). In this sense, adding another label smoothing regularizer on top of the vanilla Mixup means that the Mixup will
>  have two layers of label smoothing regularizers, which can easily mess up the regularization effect, thus degrading the accuracy
>  as shown in the results above.
>
> 2. We are glad that you found Figure 3 promising. These figures indeed show that the dynamic soft labels were evolving to adapt to the states of the learning.
> 3. We agree that our method performs on par with the SOTA method. But, our approach excludes the need to compute all the different ratios between [0, 1] for each sample pair, which is a significant reduction in terms of computation cost. Each of these mixing ratios needs to solve an optimization problem in Puzzle Mix, which is computational expensive.

---

> ### Author Response · Authors · 2020-11-24
> **Updated PDF with a set of additional results in Section6 of page 9.**
>
> We sincerely appreciate your insightful comments on our paper. We have updated the revision PDF with a set of additional results in Section 6 of page 9.
>
> Since the second discussion phase will end soon, please let us know if you have any comments/concerns that we have not addressed up to your satisfactory. We will be happy to address them to strengthen our paper.

---

### Official Review · AnonReviewer4 · 2020-10-28
**Seemingly plausible but weak support**

**Rating:** 4
**Confidence:** 4

**Review:**

Summary:
The previous advanced Mixup methods, such as CutMix and PuzzleMix, involve input mixing. This paper suggests a new Mixup approach, called LaMix, that does not require input mixing. The solution is combining the original target label (interpolation of two one-hot targets) and generated target labels from an additional network to use it for training. The authors argue that LaMix achieves superior performance without input mixing.

Reasons for score:
The authors should explain the reason why using global soft labels is theoretically plausible before describing the method. Although empirical results look nice, I am suspicious about the experimental settings for the comparison with other methods.

Pros:
- The paper includes diverse results on probing experiments.
- The paper is clearly written.

Concerns/Questions:
- As far as I understood, neural network parameters for the global soft label (W_t) are not trained because they are only used for training labels. Then, these parameters are just randomly initialized values. Is it right? If so, isn’t the final effect sensitive to the initialization?
- Sigmoid activation is used in equation (9), different from equation (7). However, there is no explanation about the reason for using it. My guess is to make artificial labels similar to each other.
- I don’t understand why beta in LaMix is 1.0 for Section 3.2 experiments. First, I think beta = 1.0 means not using the global soft label, and it is equivalent to standard Mixup. Second, the authors mention that setting beta to 0.5 is a good choice in Section 3.1.3 (Figure 4). Moreover, they use beta as 0.5 for the experiments of Section 3.1.1 (Table 1). The settings of Table 1 and Table 3 (model architectures and datasets) are the same except for the beta value of LaMix.
- Could you provide hyperparameters used for other Mixup methods as a baseline? Are they well-tuned?
- I am curious whether the combination with the global soft label can be done after input mixing. If then, I think providing these results would be helpful to check whether the regularization effect is orthogonal to input mixing.
- To compare LaMix with label smoothing, I think the author should apply label smoothing to Mixup rather than the vanilla setting.

Minor comment:
- “Puzzle Mix” -> “PuzzleMix”

---

> ### Author Response · Authors · 2020-11-21
> **Applying label smoothing to Mixup degrades accuracy.**
>
> Thank you for your insightful comments.
>
> The motivation of using the global soft labels is that there is a dis-matching issue between mixed input and associated label of a synthetic sample in Mixup. Learning the soft label helps mitigate this issue. In a nutshell, our work is motivated by two problems in Mixup. First, there is a dis-matching between mixed input and associated label in Mixup due to its random input mixing policy. Second, the input mixing for learning to mixing Mixup strategies incurs significant computation cost for selecting descriptive input regions. To cope with these two issues in Mixup, our method leverages only one mixed input (by averaging the input pair), and then assigns adaptive soft label to the same mixed input.
>
> Regarding your questions:
>
> 1. This is incorrect. W_{t} will be learned (along with the other hyperparameters of the network), before feeding into the Sigmoid, to generate the global soft label. We will make it much clearer in the revision.
> 2. In Equation9, we use a Sigmoid function, so that the output range is [0, 1]. This will provide the probability of associating the mixed input with a particular label. On the other hand, Equation7 is used by the network to generate predicted classification distribution, so there is no need for a Sigmoid to squeeze the scores into the range of [0, 1].
> 3. We found that for the random input mixing methods 0.5 provided better results. For the learning to mixing method 1.0 gave better accuracy. These two numbers were heuristically found using the validation datasets. Note that, even with Beta=1.0, our method still differs from Mixup, which uses a random mixing ratio between [0, 1] to generate a set of synthetic inputs while our method uses only one mixed input for any sample pair.
> 4. The hyperparameters are exactly the same as that in the Facebook codes and the Puzzle Mix codes as published on their websites. We didn’t change anything for fair comparison.
> 5. Thank you for suggesting applying label smoothing to Mixup. We conducted experiments of stacking uniform label smoothing (with different coefficients) on top of Mixup, and here are the results (error rates) of using PreAct ResNet-18 and ResNet-50 on Tiny ImageNet, Cifar100 and Cifar10 (number in the bracket is the deviation of 5 runs).
>
>  Tiny ImageNet:
>
>  LaMix, Mixup,  Mixup + ULS 0.1, Mixup + ULS 0.2, Mixup + ULS 0.3, Mixup + ULS 0.4
>
>  PreAct ResNet18, 34.40 (0.26), 41.68 (0.35),  42.31 (0.34), 45.19 (0.68), 60.58 (0.59), 78.48 (0.53)
>
>  ResNet50, 	 28.85 (0.15), 39.09 (0.28), 42.58 (0.39), 59.25 (0.91), 81.06 (0.24), 91.82 (0.37)
>
>  Cifar100:
>
>  LaMix, Mixup,  Mixup + ULS 0.1, Mixup + ULS 0.2, Mixup + ULS 0.3, Mixup + ULS 0.4
>
>  PreAct ResNet18, 19.38 (0.16), 21.10 (0.21), 21.51 (0.51), 21.41 (0.55), 20.94 (0.49), 20.95 (0.49)
>
>  ResNet50, 	 18.60 (0.69), 19.48 (0.48), 21.58 (0.86), 20.87 (0.51), 21.64 (0.41), 21.18 (0.58)
>
>  Cifar10:
>
>  LaMix, Mixup, Mixup + ULS 0.1, Mixup + ULS 0.2, Mixup + ULS 0.3, Mixup + ULS 0.4
>
>  PreAct ResNet18, 3.82 (0.18), 3.88 (0.32), 4.00 (0.17), 3.95 (0.13), 4.06 (0.04), 4.06 (0.03)
>
>  ResNet50, 	 3.60 (0.24), 4.29 (0.28), 4.02 (0.27), 4.09 (0.10), 4.19 (0.18), 4.68 (0.79)
>
>  These results show that label smoothing degraded the accuracy of Mixup in the cases of Cifar100, Cifar10, and Tiny ImageNet on
>  PreAct ResNet18 and Resnet50. Such accuracy degradation is due to the two layers of label smoothing reguarlization in Mixup.
>  Research has shown that the mixing label in the original Mixup has the effect of label smoothing (Carratino et al, 2020, On Mixup
>  Regularization). In this sense, adding another label smoothing regularizer on top of the vanilla Mixup means that the Mixup will
>  have two layers of label smoothing regularizers, which can easily mess up the regularization effect, thus degrading the accuracy as shown in the results above.

---

> ### Author Response · Authors · 2020-11-24
> **Updated PDF with a set of additional results in Section6 of page 9.**
>
> We sincerely appreciate your insightful comments on our paper. We have updated the revision PDF with a set of additional results in Section 6 of page 9.
>
> Since the second discussion phase will end soon, please let us know if you have any comments/concerns that we have not addressed up to your satisfactory. We will be happy to address them to strengthen our paper.

---

### Official Review · AnonReviewer3 · 2020-10-28
**An incremental work on mixup data augmentation**

**Rating:** 4
**Confidence:** 3

**Review:**

**Main Claim:**

The authors propose to use soft labels on naive mixup as an alternative to sophisticated mixup strategy. In experiments on 5 small datasets, the proposed method can achieve better accuracy than baselines.

**Strong points:**

The authors propose to use soft labels to overcome the mislearning features in mixup images. The idea is simple and straightforward.

Experiment results show the method works well on small datasets.


**Weak points:**

The contribution of this work is incremental.

Experiment results on ImageNet are missing. On Tiny-ImageNet, it’s also helpful to show the performance of Mixup and CutMix.

Some decisions are made without justification. Some details are not clearly explained. See questions.



**Recommendation:**

Reject.

The proposed method is incremental. The method should be evaluated on ImageNet.

**Questions:**

In Eq(9), a sigmoid function is applied before the softmax function. So the unnormalized logits of this distribution is in [-1, 1]. Why?

Why is “target soft labels too far away” an issue for the model? why does “a mini-batch with original inputs“ prevent “LaMix from assigning target soft labels too far away”? If the trick is not applied, what will happen to the model? Will the model take more time to converge or it won't converge?

“For LaMix, the added fully connected layer is just a copy of the fully-connected layer of the original network with a Softmax function on the top.” Is the network pre-trained? Are the two matrices sharing weights?

As shown on Figure 2, the top 2 classes occupy a large portion of the probability. So I’m curious which part actually contributes to the improvements. Is it (A) the reweighting of y_i and y_j, or (2) the introduction of other labels? I.e. After computing Eq (10), keep the value for y_i and y_j, set all other dimensions to zero, (then renormalize the distribution), and use it as the training label, what will happen?  My impression is it may solve the problem of  “target soft labels too far away”.

**Comments:**

Eq (3) is confusing to me. I think authors can follow the convention in (Yun et al.), rewrite the equation as $x_\lambda^{i,j}=\Phi(x_i, x_j, lambda) * x_i + (1 - \Phi(x_i, x_j, lambda) * x_j$

has tow forms -> has two forms

**After rebuttal:**

Thanks to the author for providing additional experimental data. But without the results of imagenet, it is difficult to judge the effectiveness of this method on complicated data. So I decided to keep the original score.

---

> ### Author Response · Authors · 2020-11-21
> **Added ablation studies and additional experiments as suggested.**
>
> Thank you for your insightful comments.
>
> In terms of contribution, our research shows that one can bypass the random input mixing in Mixup to conduct sufficient model regularization, and we think this message is worth noting to the research community.
>
> In terms of our technical novelty, the use of adaptive soft label in our work is motivated by the mis-matching issue between a mixed input and its soft label in Mixup. This is due to the fact that Mixup generates soft labels using random mixing ratios for linear interpolation. Such mis-matching issue will result in generating misleading synthetic samples and causing underfitting. To cope with this challenge, our method fixes the mixed input (by averaging the input pair), and then adaptively assigns a soft label to the same mixed input. The learned soft label aims to mitigate the potentially wrong soft label generated by Mixup.
>
> We conducted additional experiments on Tiny-ImageNet. Here is the results for Resnet-18 and ResNet-50. We will add the following results (error rates) to Table 4.
>
>  Vanilla, Mixup, CutMix
>
>  PreAct ResNet18,  42.61 (0.41), 41.68 (0.35), 43.01 (0.36)
>
>  ResNet50, 	  40.89 (0.36), 39.09 (0.28), 41.36 (0.44)
>
>
> We also agree that results on ImageNet will improve our paper, but experiments on ImageNet are just too expensive to us. We conducted additional experiments on Tiny ImageNet and will add the following results (error rates) into Table 1.
>
>
> Vanilla, Mixup, CutMix, LaMix
>
>  PreAct ResNet18,  42.61 (0.41), 41.68 (0.35), 43.01 (0.36), 34.40 (0.26)
>
> ResNet50, 	  40.89 (0.36), 39.09 (0.28), 41.36 (0.44), 28.85 (0.15)
>
>
> Regarding your Questions:
>
> 1. We use a Sigmoid function in Eq(9), so the output range is [0, 1] (not [-1, 1]). The reason here is that this function will provide the probability of associating the input to a particular label.
> 2. Without the original samples, the method significantly degraded its accuracy. We conducted the suggested experiments and provided the results (error rates) as follows (average over five runs and their standard deviations in brackets). The reason here is that, without the original training samples, the training samples could be very different from the testing samples.
> Also, it is interesting to see that ResNet50 obtained inferior results than PreAct ResNet18 in some cases when removing the original training samples. This indeed suggests that Resnet50 fits the mixed training samples better due to its expressive power, but those mixed training samples may be far from the testing samples due to the lack of the original training samples.
>
>  Cifar100:
>
>  LaMix, LaMix w/o original samples
>
>  PreAct ResNet18, 19.38 (0.16), 25.74 (0.43)
>
>  ResNet50, 	 18.60 (0.69), 25.57 (0.36)
>
>  Cifar10:
>
>  LaMix, LaMix w/o original samples
>
>  PreAct ResNet18, 3.82 (0.18), 5.90 (0.31)
>
>  ResNet50, 	 3.60 (0.24), 6.39 (0.59)
>
>
> 3. By “the added fully connected layer is just a copy of the fully-connected layer” we meant that the structure of the two are the same, namely W_{t} and W_{l}. The networks are trained end-to-end simultaneously. They are two different matrices and they don’t share weights. We will make it much clearer in the revision.
> 4. The contributions of the performance improvement come from both: mixing of y_{i} and y_{j} and the learned smoothing distribution P^{ij}. Without the p^{ij}, the model significantly degraded the accuracy. We conducted the experiments as suggested and provided the results (error rates) below (average over five runs with standard deviation in the bracket), which show that the model degraded the accuracy:
>
>  Cifar100:
>
>  LaMix, LaMix w/o global soft label
>
>  PreAct ResNet10,  19.38 (0.16), 23.03 (0.35)
>
>  ResNet50, 	  18.60 (0.69), 21.34 (0.70)
>
>  Cifar10:
>
>  LaMix, LaMix w/o global soft label
>
>  PreAct ResNet10, 3.82 (0.18), 5.07 (0.38)
>
>  ResNet50, 	 3.60 (0.24), 4.78 (0.14)
>
>
> 5. Thank you for suggesting the equation from Yun et al. The suggested formula will not cover the case of Puzzle Mix due to the optimal transport component in  Puzzle Mix.

---

> ### Author Response · Authors · 2020-11-24
> **Updated PDF with a set of additional results in Section6 of page 9.**
>
> We sincerely appreciate your insightful comments on our paper. We have updated the revision PDF with a set of additional results in Section 6 of page 9.
>
> Since the second discussion phase will end soon, please let us know if you have any comments/concerns that we have not addressed up to your satisfactory. We will be happy to address them to strengthen our paper.

---

### Official Review · AnonReviewer1 · 2020-10-29
**Label smoothing in mixup**

**Rating:** 4
**Confidence:** 4

**Review:**

This work proposes to change the labels for the mixed examples in mixup. My major concerns are as follows.
1.	The motivation of adopting soft label for mixup is not clear. Label smoothing is helpful for generic training but why it can benefit mixup?
2.	The proposed method is more like a combination of mixup and label smoothing. The improvement may come from label smoothing as a generic trick rather than mixup itself.
3.	The performance of proposed method is very close to mixup, where the improvement is not significant. Additional experiments on ImageNet can make the results more convincing.

---

> ### Author Response · Authors · 2020-11-21
> **Mixup plus label smoothing won't work.**
>
> Thank you for your insightful comments.
>
> 1. The use of adaptive soft label in our work is motivated by the mis-matching issue between a mixed input and its soft label in Mixup. This is due to the fact that Mixup generates soft labels using random mixing ratios for linear interpolation. Such mis-matching issue will result in generating misleading synthetic samples and causing underfitting. To cope with this challenge, our method fixes the mixed input (by averaging the input pair), and then adaptively assigns a soft label to the only mixed input. Dynamically learning a soft label will mitigate the label dis-matching issue in Mixup and also effectively make good use of the only mixed input. On the contribution side, we also think that the message of bypassing the random input mixing in Mixup is worth noting to the community.
>
>
> 2. Our research is motivated by the label and input mis-matching issue in Mixup and our solution is inspired by label smoothing and self-distillation. In fact, simply stacking label smoothing on top of vanilla Mixup didn’t help. We conducted experiments of stacking uniform label smoothing (with different coefficients) on top of Mixup, and here are the error rates of using PreAct ResNet-18 and ResNet-50 on Tiny ImageNet, Cifar100 and Cifar10 (number in the bracket is the deviation of 5 runs).
>
>  Tiny ImageNet:
>
>  LaMix, Mixup,  Mixup + ULS 0.1, Mixup + ULS 0.2, Mixup + ULS 0.3, Mixup + ULS 0.4
>
>  PreAct ResNet18, 34.40 (0.26), 41.68 (0.35), 42.31 (0.34), 45.19 (0.68), 60.58 (0.59), 78.48 (0.53)
>
>  ResNet50, 	 28.85 (0.15), 39.09 (0.28), 42.58 (0.39), 59.25 (0.91), 81.06 (0.24), 91.82 (0.37)
>
>  Cifar100:
>
>  LaMix, Mixup,  Mixup + ULS 0.1, Mixup + ULS 0.2, Mixup + ULS 0.3, Mixup + ULS 0.4
>
>  PreAct ResNet18, 19.38 (0.16), 21.10 (0.21), 21.51 (0.51), 21.41 (0.55), 20.94 (0.49), 20.95 (0.49)
>
>  ResNet50,  18.60 (0.69), 19.48 (0.48), 21.58 (0.86), 20.87 (0.51), 21.64 (0.41), 21.18 (0.58)
>
>  Cifar10:
>
>  LaMix, Mixup, Mixup + ULS 0.1, Mixup + ULS 0.2, Mixup + ULS 0.3, Mixup + ULS 0.4
>
>  PreAct ResNet18, 3.82 (0.18), 3.88 (0.32), 4.00 (0.17), 3.95 (0.13), 4.06 (0.04), 4.06 (0.03)
>
>  ResNet50, 	 3.60 (0.24), 4.29 (0.28), 4.02 (0.27), 4.09 (0.10), 4.19 (0.18), 4.68 (0.79)
>
>  These results show that label smoothing degraded the accuracy of vanilla Mixup in the cases of Cifar100, Cifar10, and Tiny
>  ImageNet. Such accuracy degradation is due to the two layers of label smoothing in Mixup. Research has shown that the mixing
>  label in the original Mixup has the effect of label smoothing (Carratino et al, 2020, On Mixup Regularization). In this sense, adding
>  another label smoothing regularizer on top of the vanilla Mixup means that the Mixup will have two layers of label smoothing
>  regularizer, which can easily mess up the regularization effect, thus degrading the accuracy as shown in the results above.
>
>
> 3. In terms of performance of our method, as shown in the experiments, Mixup did obtain inferior accuracy in some cases when comparing to CutMix, but our method outperformed both Mixup and CutMix in all testing cases.
> We agree that experiments on ImageNet will improve our paper, but unfortunately these experiments are too expensive to us. We conducted additional experiments on Tiny ImageNet, and the following results (error rates) will be added to Table 1 in the revision:
>
>  Vanilla, Mixup, CutMix, LaMix
>
>  PreAct ResNet18,  42.61 (0.41), 41.68 (0.35), 43.01 (0.36), 34.40 (0.26)
>
>  ResNet50, 	  40.89 (0.36), 39.09 (0.28), 41.36 (0.44), 28.85 (0.15)

---

> ### Author Response · Authors · 2020-11-24
> **Updated PDF with a set of additional results in Section6 of page 9.**
>
> We sincerely appreciate your insightful comments on our paper. We have updated the revision PDF with a set of additional results in Section 6 of page 9.
>
> Since the second discussion phase will end soon, please let us know if you have any comments/concerns that we have not addressed up to your satisfactory. We will be happy to address them to strengthen our paper.

---

> ### Author Response · Authors · 2020-11-24
> **LaMix significant outperformed Mixup on TinyImageNet (over 10% absolute error rate reduction)**
>
> As shown in Table 5 on Page 9, for the Tiny ImageNet LaMix obtained error rates of 34.40% and 28.85% with PraAct ResNet-18 and ResNet-50 respectively, which significantly outperformed the 41.68% and  39.09% obtained by Mixup.

---

### Decision · Program_Chairs · 2021-01-07
**Final Decision**

**Decision:**

Reject

**Comment:**

This work proposes to improve Mixup by using soft labels, removing the need for input mixup. The reviewers found the paper was clear and found the experiments promising. The reviewers raised concerns about the lack of experiments comparing this approach to Mixup+Label smoothing, which were addressed during the rebuttal by the authors. However, the reviewers did not find the empirical evidence strong enough given that this is mostly an empirical contribution. The authors do not necessarily need to train on the full Imagenet, but it would be beneficial to evaluate on more standard settings on the dataset considered to facilitate comparison to previous work.